# Classification with a Deferral Option and Low-Trust Filtering for Automated Seizure Detection

**DOI:** 10.3390/s21041046

**Published:** 2021-02-04

**Authors:** Thijs Becker, Kaat Vandecasteele, Christos Chatzichristos, Wim Van Paesschen, Dirk Valkenborg, Sabine Van Huffel, Maarten De Vos

**Affiliations:** 1I-Biostat, Data Science Institute, Hasselt University, 3500 Hasselt, Belgium; dirk.valkenborg@uhasselt.be; 2STADIUS Center for Dynamical Systems, Signal Processing and Data Analytics, Department of Electrical Engineering (ESAT), KU Leuven, 3001 Leuven, Belgium; kaat.vandecasteele@esat.kuleuven.be (K.V.); christos.chatzichristos@esat.kuleuven.be (C.C.); sabine.vanhuffel@esat.kuleuven.be (S.V.H.); maarten.devos@kuleuven.be (M.D.V.); 3Department of Neurology, UZ Leuven, 3001 Leuven, Belgium; wim.vanpaesschen@uzleuven.be; 4Laboratory of Epilepsy Research, KU Leuven, 3001 Leuven, Belgium; 5Department of Development and Regeneration, KU Leuven, 3001 Leuven, Belgium

**Keywords:** epilepsy, seizure detection, electroencephalography, classification with a deferral option, home monitoring, long-term monitoring, wearables

## Abstract

Wearable technology will become available and allow prolonged electroencephalography (EEG) monitoring in the home environment of patients with epilepsy. Neurologists analyse the EEG visually and annotate all seizures, which patients often under-report. Visual analysis of a 24-h EEG recording typically takes one to two hours. Reliable automated seizure detection algorithms will be crucial to reduce this analysis. We investigated such algorithms on a dataset of behind-the-ear EEG measurements. Our first aim was to develop a methodology where part of the data is deferred to a human expert, who performs perfectly, with the goal of obtaining an (almost) perfect detection sensitivity (DS). Prediction confidences are determined by temperature scaling of the classification model outputs and trust scores. A DS of approximately 90% (99%) can be achieved when deferring around 10% (40%) of the data. Perfect DS can be achieved when deferring 50% of the data. Our second contribution demonstrates that a common modelling strategy, where predictions from several short EEG segments are combined to obtain a final prediction, can be improved by filtering out untrustworthy segments with low trust scores. The false detection rate shows a relative decrease between 21% and 43%, and the DS shows a small increase or decrease.

## 1. Introduction

Epilepsy is a neurological disorder that affects around 0.8% of the population worldwide [1]. Epilepsy patients have recurrent unprovoked seizures, which significantly affect their quality of life. Anti-epileptic drugs provide adequate treatment for 70% of the patients [2]. The seizure burden is an important variable for treatment decisions and the evaluation of drug trials. One should therefore be able to objectively document seizures occurring over a time span of days to weeks [3], preferably in a home environment [4]. Unfortunately, seizure reporting by patients is unreliable [5]. Seizures are therefore detected with devices that record biosignals, most commonly full scalp electroencephalography (EEG) [6], which is uncomfortable to wear for a long period of time. Other biosignals such as electrocardiograms, electromyograms, accelerometry and EEG from behind-the-ear sensors can be used outside the hospital [7]. They have the advantage that those measuring devices are more tolerated when being worn for an extended period of time. Combining several of these biosignals can improve seizure detection performance [8,9].

Regardless of the measured biosignal(s), manual analysis of the output is a time-consuming task. Automated seizure detection with machine learning has therefore received a lot of attention [10,11,12]. Impressive results have been obtained, though mostly on retrospective single-center datasets. Average detection sensitivities (DS) between 80% and 90% have been reported [13,14,15]. However, the DS for individual patients can still be low in these models. Low DS can be problematic, especially when evaluating possibly small changes in the seizure burden, which is relevant for, e.g., evaluating the effect of a newly administered drug. Seizure detection systems with a DS that is guaranteed to be close or equal to 100% for all patients are therefore of interest.

Achieving a performance of automated seizure detection that is sufficient for implementation in a clinical setting remains challenging. Automated seizure detectors that are commercially available have been reported to have a low detection sensitivity: using full scalp EEG they detected at least one seizure in only 53% of measurements containing seizures [16]. Recently, a seizure detection competition was held on the Temple University Hospital Seizure Detection Corpus, which is the largest open source corpus of its type and includes representative cases of different types of seizures [17]. Despite the size of the dataset and the use of advanced machine learning algorithms, the participants were unable to achieve a performance that is sufficient for clinical practice and that could be used for all types of seizures [18].

We investigated a dataset containing EEG measurement from four behind-the-ear sensors, from our SeizeIT1 study [19,20]. Behind-the-ear sensors are able to detect epileptic seizures for focal onset and generalised seizure types [21,22]. In contrast to full scalp EEG, there are only a few studies that investigated automated seizure detection algorithms on behind-the-ear EEG [22,23,24].

Our first contribution in this paper, is the evaluation of the performance of the classifier in case that the EEG segments for which the classifier is least confident are deferred to a human annotator, who is assumed to annotate perfectly. A similar scenario is quite common in clinical epilepsy research: the algorithm flags all suspicious activity, which is then presented to a human annotator [25]. Learning algorithms with a reject option have a long history in machine learning research [26,27]. Classification with the option to defer to a human expert has received increasing attention in the current AI literature [28,29,30,31], and is particularly relevant for medical tasks [32,33,34]. Application of this approach to seizure detection has been limited. Computer-assisted detection of epileptic discharges from full scalp EEG has been investigated by Clarke et al. [25]. On a retrospective dataset, a neural network achieved a DS of 96.7% with 1670 false detections per 24 h. They employed this model in a clinical application of ambulatory measurement of 7 patients with idiopathic generalised epilepsy. 10-s EEG segments that contained a seizure detection were deferred to a human annotator. The data that needed to be reviewed was reduced to between 60% and 90% of the full data. The precision and false detection rate (FDR) were tuned by changing the classification threshold of the network. The DS was not measured for the clinical application.

In this article, we defer EEG segments with a length of at least 5 min to a human annotator. We investigated the support vector machine (SVM) models that were trained to perform automated seizure detection on patients with focal epilepsy in [23]. We used the confidences derived from the SVM output or from so-called trust scores [30] (which only depend on the labels and on the features derived from the EEG to train the SVM). Our second contribution demonstrates that a common modelling strategy, where predictions from several short segments are combined to obtain a final prediction [9,22,23,35], can be improved by filtering out untrustworthy segments. This filtering approach only works with the trust scores. For the SVM confidences it lowers the performance. Our results indicate that it is of interest to investigate algorithms that combine a good classification performance with good confidence estimates, instead of focusing solely on the performance when all data is classified by the algorithm. Since these results have been obtained on behind-the-ear EEG data, they have the potential to be used for long-term home monitoring of people with epilepsy.

## 2. Materials and Methods

Most code was implemented in a Conda environment in Python 3. The main libraries that were used are scikit-learn [36] and SciPy [37]. Our code is made available at https://github.com/thijsrmbecker/classify_w_deferral_seizure. Some preliminary data processing was done in Matlab.

### 2.1. Dataset

The dataset consists of recordings from a traditional 10–20 scalp EEG with four extra behind-the-ear electrodes [20,22,38], as shown in Figure 1. The neurologist (W.V.P.) annotated all seizures on the standard video-EEG recordings. In the present study, we only used the behind-the-ear EEG measurements plus the seizure annotations of the standard video-EEG recordings, and call these “full seizure events”. We only took into consideration the seizure segments that could be blindly annotated by the neurologist (W.V.P.) in the behind-the-ear measurements, which are 63% of all seizures. These included mainly patients with focal seizures from temporal lobe origin or patients with other focal epilepsy syndromes with ictal propagation that was picked up by the behind-the-ear electrodes, and two patients with focal to bilateral tonic clonic seizures. 54 patients are included, 42 of which had seizures during the measurements. The dataset consists of approximately 220 days of EEG time series and contains 114 seizures. The average measurement time per patient is 4.1 days with a standard deviation of 0.9 days. The median measurement time is 4.2 days. The minimum recording time is 0.74 days and the maximum recording time is 4.9 days. We refer to [23] for a detailed discussion of the content of the dataset. The investigated dataset is from our SeizeIT1 study [19,20]. The equipment was manufactured by OSG-EEG from Rumst, Belgium. We are currently performing the SeizeIT2 study, which is a multicenter study to examine clinical scenarios for long-term monitoring of epileptic seizures with a wearable biopotential technology in the home environment [39] (ClinicalTrials.gov Identifier: NCT04284072).

### 2.2. Models

We investigated the patient-independent SVMs presented in [23]. The goal of the models is to detect focal seizures visible on the behind-the-ear EEG by the neurologists, and with a length of at least 10 s. Features were extracted from 2-s segments with 50% overlap. Seizures segments have label 1 and seizure-free segments have label 0. If more than 7 out of 10 subsequent 2-s segments are classified as a seizure by the SVM, the 10-s segment is classified as a seizure (also referred to as a seizure flag). If there are subsequent seizure flags, only the first flag is retained.

For each seizure a 10-s EEG segment that contains a clear ictal EEG pattern was selected. More specifically, the first 10 s with a typical ictal pattern were always selected. Those patterns were identified by an increased amplitude compared with the background. If there was no increase in amplitude, the first 10 s with a change in frequency content were selected. If the whole seizure contained artifacts or no typical ictal patterns were observed, the first 10 s after the seizure onset were selected for training. The SVM model from [23] was trained on these 10-s segments with a clear ictal EEG pattern. This procedure ensured that each seizure has the same importance when training the SVM, independent of its total length. An example of the full seizure labels and the clear ictal labels is shown in Figure 2. The SVM trained on the clear ictal segments has a FDR per 24 h of 2.9 and a DS of 64.1%. It is referred to as clear ictal SVM (CI SVM). We also investigated an SVM model trained on the full seizure labels. It has a FDR per 24 h of 17.2 and a DS of 83.0%. It is referred to as full seizure SVM (FS SVM).

### 2.3. Confidence Measures

The confidence in the prediction of the SVMs was derived from the distance *d* to their separating hyper-plane. Temperature scaling [40] was performed to convert these distances to probabilities:(1)$$p_{a}=\frac{1}{1+\exp\left(-a.d\right)},$$
where a>0 is a parameter which can be optimised. Temperature scaling is a simplified version of Platt scaling [41], which is often used to obtain probabilities from SVM outputs. In contrast to Platt scaling, the class of the predictions cannot change with temperature scaling. Temperature scaling is surprisingly effective at calibrating neural networks [40]. As the classification threshold is at $p_{a}=0.5$ (i.e., d=0), the confidence is calculated as $|p_{a}-0.5|$.

Trust scores were introduced in [30]. They were calculated with the code from https://github.com/google/TrustScore. The algorithm consists of 2 steps. In the first step one calculates the α-high-density set of each class. This is done by removing the α-fraction of the samples with the lowest density of each class (which may be outliers). The samples in the α-high-density set can be interpreted as the "representative" fraction of each class. α is a variable determined by the user, which can be optimised. In the second step, the trust score of each point in the test set is found by calculating its (Euclidean) distance to the closest point of the α-high-density set of the other class, and dividing it by the distance to the closest point of the α-high-density set of the predicted class.

We skipped the first step when calculating the trust scores. This has two important advantages: It makes the calculation significantly faster; and it eliminates the only 2 hyper-parameters associated with the trust model (α and a parameter *k* to estimate the empirical density based on *k*-nearest neighbours). The disadvantage is that we could lose some performance. The original 67 features extracted from the EEG time series [23] were reduced to 20 dimensions using principal component analysis. We did not optimize for the number of principal components. We took this number because it was also used in the original article on trust scores [30].

A low trust score can be interpreted as a sample that is atypical for its predicted class. This could be a result of the presence of noise or artifacts in the EEG, but could also be caused by other reasons. Under some distributional assumptions one can show that a high (low) trust score implies that the classifier likely agrees (disagrees) with the Bayes-optimal classifier [30]. As trust scores are independent of the classifier, they can be used in conjunction with any classifier. We trained the trust models using either the full seizure or clear ictal labels.

Trust models were fitted on a subset of the data. All seizure segments were included. For each patient, one-minute long non-seizure segments were selected 15 min apart, with each segment containing 30 non-overlapping 2-s segments. Out of all these selected one-minute long segments, we randomly drew 100 for each patient. If there was not enough data to randomly draw 100 segments, the one-minute segments were selected 5 min apart. This procedure was done to assure that fitting the trust models and calculating the trust scores was fast enough.

### 2.4. Performance Metrics

If a seizure flag occurs between the onset and the end of a seizure, it counts as a true positive (TP). If no seizure flags occur between the onset and the end of a seizure it is a false negative (FN). All seizure flags that do not overlap with a seizure are false positives (FP). The performance metrics were calculated as in [23]: detection sensitivity = TP/(TP + FN); false detection rate = FP/recording length, where FP within 10 s of each other are counted as one FP; positive predictive value (PPV) = TP/(TP + FP) (also called precision); and F1-score = 2 TP/(2 TP + FP + FN). The PPV and F1-score were only calculated on patients that have seizures. The detection delay is the time difference in seconds between the start of the seizure and the seizure flag. All performance metrics were calculated per patient and then averaged.

### 2.5. Classification with a Deferral Option

If a segment is annotated by a human, we assumed it is done perfectly. The full EEG signal was divided into segments that can be deferred to a human annotator as follows. All 10-s seizure flags were put in the middle of a 5-min segment. If there is less than 5 min between two such 5-min segments, they were merged. Afterwards, the remaining EEG signal (that does not contain seizure flags) was divided into 5-min segments. Segments that contain a seizure flag are always deferred. Other segments are deferred based on their confidence score. The deferral strategy is visualised in Figure 3.

The confidence score of each segment was calculated as follows. We considered the trust scores or SVM confidences (referred to as score in this Section) of all 2-s segments in the segment. We either calculated the average of the scores of all 2-s segments, or we took the percentage $p_{\mathrm{low}}$ of 2-s segments with the lowest score and calculated the average of only those 2-s segments. $p_{\mathrm{low}}$ is a hyper-parameter that can be optimised.

The same percentage of lowest-confident segments is deferred for each patient. After deferring these segments, remaining adjacent segments are merged. If a seizure is in a deferred segment for at least 10 s, it counts as detected; the classification of the part (if any) that is annotated by the algorithm does not influence the performance. If an undetected seizure is split in such a way that both the deferred and non-deferred part contain less than 10 s, it counts as undetected.

### 2.6. Low-Trust Filtering

#### 2.6.1. General Procedure

From now on, we refer to 2-s segments simply as segments while all the other segment lengths will always be explicitly specified. In the original algorithm [23], a 10-s EEG segment is classified as a seizure if more than 7 out of 10 of the segments are classified as a seizure. This classification is influenced by segments that are difficult to classify (noise, artifacts, etc.) and could, potentially, negatively influence the performance. Since we have 10 predictions, we can remove these “untrustworthy” predictions while still classifying all 10-s segments. We call this approach low-trust filtering (LTF). The new classification rule is as follows: If less than 5 out of 10 predictions are removed, a seizure flag is created if the mean of the remaining predictions is greater than 0.7; otherwise, a seizure flag is created if the mean of the 5 highest-trusted predictions is greater than 0.7. This classification rule can be seen as a generalisation of the rule from [23], with the requirement that at least 5 predictions should be taken into account. We did not optimize for this classification rule, but tried out two variations of this rule in early experiments. These other two classification rules and the arguments for our final choice are discussed in the Appendix A.

A visualisation of how LTF works is shown in Figure 4. In this Figure, a false detection is removed by LTF. In the orange band, 8 out of 10 predictions are 1. According to the classification rule before LTF, it is therefore classified as a seizure. However, 7 of the seizure predictions were deemed untrustworthy. Following the classification rule for LTF, we took the 5 highest-trusted predictions, which are 2 non-seizure and 3 seizure predictions. The average prediction value is 0.6, which is lower than 0.7, so the segment is no longer predicted to be a seizure.

#### 2.6.2. Cross-Validation Approach

Cross-validation was performed with the F1-score. Both parameters (*a* and the percentage of segments to filter) were selected through a grid search approach. The same percentage of segments was filtered for each patient.

For LTF with trust scores, the only parameter that needs to be defined is the optimal percentage of lowest-trust segments to filter. This has to be done on a validation set containing patients which were not used to train the trust model. To avoid over-fitting, we performed nested cross-validation [42]. This is a conservative approach to cross-validation [43], so we did not expect any possibility of over-fitting. The 54 patients were divided into 6 folds containing 9 patients each. The total dataset contains approximately 220 days of EEG time series, with a total of 490 10-s seizure segments (obtained from 114 seizure events). Folds were randomly created, with the constraints that each fold should contain between 65 and 96 10-s seizure segments, and between 23.1 and 48.6 days of EEG data. The cross-validation approach is visualised in Figure 5. Each fold was used as the test set once. It is the performance on the test set that is reported. We can now explain the procedure when the test set contained the patients of the first fold. We merged folds 2 and 3 (fold${}_{23}$), and folds 4 and 5 (fold${}_{45}$). Fold 6 was split in half and each half was merged with fold${}_{23}$ or fold${}_{45}$. The end result was one fold containing 22 patients and another fold containing 23 patients, with a good balance of number of seizures and amount of EEG data between the folds. We performed 2-fold cross-validation with fold${}_{23}$ and fold${}_{45}$: we fit a trust model on the patients in fold${}_{23}$ and fold${}_{45}$, and determined the optimal percentage to filter from fold${}_{45}$ and fold${}_{23}$, respectively. The average of these two optimal percentages to filter was used to determine the performance on the test fold.

After using cross-validation to determine the optimal percentage of lowest-trust segments to filter, we still had to pick a model that determines which segments to filter. To achieve optimal performance, this was done with a trust model fitted on all 53 patients besides the patient under consideration.

For LTF with the SVM confidence scores we used leave-one-patient-out cross-validation: the optimal parameters were determined for all patients except the patient under consideration. We optimised with both the lowest-trust percentage to filter and the *a* variable from Equation (Equation 1). Taking the distances to the separating hyper-plane *d* without performing temperature scaling was also considered.

#### 2.6.3. Significance Testing

We tested the null hypothesis that the distribution of the performances is the same with the Wilcoxon signed-rank test, which is a paired difference test. We either compared the distribution of a performance metric (DS, PPV, F1-score, etc.) between the model without LTF and with LTF, or we compared the model with the highest value for a given metric with all the other models. As we always tested whether a model had a higher or lower performance compared to the reference model, we used a one-sided test. We rejected the null hypothesis if the p-value was lower than 0.05.

## 3. Results

The main results are discussed here. Additional results are presented in the Appendix A.

### 3.1. Classification with a Deferral Option

For the SVM confidences we used the distances to the separating hyper-plane. These provided the same results as temperature scaling.

An example of the behaviour of the DS for different $p_{\mathrm{low}}$ is shown in Figure 6. The optimal $p_{\mathrm{low}}$ lies between 1 and 10. $p_{\mathrm{low}}=100$ is the worst performance possible. From now on, we report results for $p_{\mathrm{low}}=5$.

The results for the CI SVM are shown in Figure 7. After deferring all segments that contain a seizure flag, which is approximately 1% of the data, one gets a FDR of 0. A DS of 89% (99%) was reached after deferring 11% (36%) of the data. Perfect performance on all considered metrics was reached when deferring 50% of the data. SVM confidences slightly outperformed trust scores.

The results for the FS SVM are shown in Figure 8. After deferring all segments that contained a seizure flag, which was approximately 4.5% of the data, one got an FDR of 0. A DS of 90% (99%) was reached after deferring 9% (38%) of the data. In this case, trust scores slightly outperformed SVM confidences. These results are very similar to the CI SVM, despite the difference in performance when no data is deferred. Perfect performance on all considered metrics was reached when deferring 62% of the data. This is worse compared to the CI SVM. However, the improvement from 99% to 100% DS is determined by the detection of one seizure. The exact point at which 100% DS occurs is expected to be subject to quite some random variation. See, e.g., the variation in DS for different $p_{\mathrm{low}}$ in Figure 6.

The number and average length of the deferred segments as functions of the fraction of deferred data are plotted for the CI SVM in Figure 9. They were normalised per patient and per 24 h of EEG data. The average length of a deferred segment starts at around 5 min for deferral percentages close to 0, as expected. As adjacent deferred segments are merged, the average segment length increases for higher deferral percentages. At 10% deferral, around 20 segments with an average length of 10 min are deferred. At 40% deferral we defer around 40 segments with an average length of 15 min. The maximum number of deferred segments is approximately 45 and is reached at around 50% deferral. The behaviour of the FS SVM is similar, as shown in the Appendix A.

### 3.2. Low-Trust Filtering

We first discuss the results for the CI SVM; see Table 1 for the average performance. If one performs LTF with the trust models fitted on the CI labels, the DS decreases slightly but non-significantly. The FDR per 24 h decreases from 2.9 to 1.7, a relative decrease of 41%. The PPV goes from 38.9% to 48.6%, and the F1-score from 39.7% to 49.2%. If the FS labels are used to fit the trust models, LTF lowers the FDR and increases the DS. Both improvements are significant. The PPV is 50.8% and the F1-score is 52.1%. LTF with the CI SVM confidences did not improve the performance, and in fact degraded the FDR. The distances *d* and temperature scaling optimised for *a* provided the same results. On average, 2% of the lowest-trust segments are filtered for LTF with the trust models trained on the FS labels. Similar behaviour was observed for the median performances, see Table 2.

The results for the FS SVM are given in Table 3 for the average performance. For the trust models trained on the FS labels, the FDR per 24 h decreases from 17.2 to 10.6, a relative decrease of 38%. The DS decreases slightly but non-significantly. The PPV goes from 12.6% to 20.3% and the F1-score goes from 18.3% to 27.7%. LTF with the trust models trained on the CI labels performs slightly better. LTF with the FS SVM confidences did not improve the performance, and in fact degraded the FDR. The distances *d* and temperature scaling optimised for *a* gave the same results. On average, 10% of the lowest-trust segments are filtered for LTF with the trust models trained on CI labels. Similar behaviour was observed for the median performance, see Table 4.

The LTF results are similar for both SVM models. LTF with trust models led to a relative decrease of the FDR of around 40% for three out of the four cases. For those cases a small decrease of the DS was observed. In the other case (CI SVM with a trust model trained on the FS labels) the FDR showed a relative decrease of approximately 20%, with a small increase of the DS. The strong decrease in FDR led to strong improvements in the PPV and F1-score. No pronounced effects were observed for the detection delays. In both cases, LTF with the trust models trained on the labels different from the labels used to train the SVM slightly outperformed the other trust model. SVM confidences did not improve the performance, and in fact increased the FDR.

Figure 4 shows a visualisation of a false positive segment that was removed after performing LTF. In this seizure-free EEG signal, there is a part where the SVM classified many segments as seizures. All but one of these seizure predictions were deemed untrustworthy by the trust scores, so the seizure flag disappeared after LTF.

### 3.3. Difference between SVM Confidences and Trust Scores

We compare the rankings of the SVM confidences and the trust scores with the Kendall rank correlation coefficient (KRCC). The KRCC was calculated between the SVM confidences and trust scores for each patient and then averaged. For the CI SVM confidences, the KRCC was 0.16 with the trust model trained on the CI labels, and 0.17 with trust model trained on the FS labels. For the FS SVM confidences, the KRCC was 0.16 with the trust model trained on the FS labels, and 0.11 with the trust model trained on the CI labels. These KRCC values are low, and therefore show that the rankings of the SVM confidences and trust scores differ notably.

Trust scores are superior for removing false positives for both LTF and classification with a deferral option. This can be seen in Table 1, Table 2, Table 3 and Table 4 for LTF. For classification with a deferral option, this is shown in Figure 10. In this figure we plot the FDR as a function of the fraction of the data that is deferred to a human annotator. In contrast to our main approach, the segments that contain seizure flags are not automatically the first to be deferred. If a seizure flag is in a deferred segment for at least one second, we assumed that it is completely checked by the human annotator, even if a part is checked by the algorithm. It is clear that trust scores are better at detecting segments that contain false positives. For detecting 5-min segments that contain false negatives our results suggest that SVM confidences and trust scores behave similarly. Either SVM confidences (CI SVM, Figure 7) or trust scores (FS SVM, Figure 8) performed better. However, the difference in DS is determined by only a few seizures, so these results are quite noisy, and one should not draw strong conclusions based on these results.

## 4. Discussion

To appreciate how well deferring segments to a human annotator works, note that for the baseline strategy of deferring random segments the average performance stays constant and becomes more noisy. Compared to this scenario the scaling of the performances is good: the average performances increase monotonically and the standard deviations decrease. We investigated some basic deferral strategies: taking the average of the least-confident percentage $p_{\mathrm{low}}$ of the 2-s segments in the 5-min segments; using SVM confidences or trust scores; and taking the same or a different percentage of the data to defer per patient (discussed in the Appendix A). These strategies already showed significant variations in performance. Perfect performance can be reached when deferring 50% of the data. Our results indicate that more sophisticated deferral schemes could further improve our results. In our application, the SVM is not aware that it can defer data. Which data to defer is decided after training the SVM. One can train a classifier and a rejector simultaneously [28,32,33,34], which can improve both the classifier and the deferral decisions. Our current strategy consists of selecting 5-min segments and taking the average of the confidences of the 2-s segments included in this 5-min segment. Making the lengths of the segments variable and making the deferral choice based on more detailed properties than the average confidence could be beneficial. Better classification models and better confidence estimates are also relevant research directions. Instead of deferring EEG segments that do not contain any detected seizures, one could also choose to increase the threshold of the classifier until 100% DS is reached [44]. Finally, more or multi-modal data would likely improve the performance.

To achieve clinical applicability, the amount of data that needs to be deferred to achieve perfect detection sensitivity should be reduced to around 5% to 10%. In that case, such models could be implemented, and medical specialists would only need to manually review a small amount of data while still achieving perfect performance. Once implemented, the continuously collected data could be used to further train the model, which would further reduce the percentage of data needing to be deferred.

For the investigated dataset, LTF was a valuable strategy to improve the performance of an already trained classifier. It is relatively straightforward to implement, since only the “percentage of segments to filter” hyper-parameter needs to be optimised. If one fits a model and afterwards performs LTF, the likelihood of introducing over-fitting is therefore small. In the nested cross-validation scheme of this article, the probability of over-fitting is negligible. Calculating the trust score for a 2-s segment takes 0.0016 s on a laptop (dual core i5-6200 CPU with 16 GB RAM), so LTF could be applied in online seizure detection. A mechanism similar to LTF could be achieved by attention networks [45]. These are neural networks that learn to weight the input, depending on its relevance. Attention networks have already been applied to seizure detection [18].

Note that for both classification with a deferral option and LTF, we did not remove segments by determining whether they were below a certain numerical threshold; rather, we removed the lowest-trusted percentages. We are therefore interested in a good uncertainty ranking of the predictions, which is not necessarily the same as a good calibration. A well-calibrated model returns probabilities that reflect the likelihoods of its predictions. Deep neural networks seem to return good uncertainty rankings [30,40], even though they are not necessarily well calibrated [40]. There are empirical studies on what methods are optimal for calibrating machine learning classifiers [46,47]. SVMs that are calibrated by Platt scaling perform moderately well, whereas random forests and small neural networks perform better. It is, however, unclear if the results from such studies can be directly translated to seizure detection, for which there is a very large class imbalance. There is, furthermore, evidence that the optimal calibration method can change depending on the dataset. It was found that datasets with labels that contain some inherent uncertainty (i.e., where experts would disagree on some labels) benefit from different calibration methods compared to datasets without that extra uncertainty [48]. Given these issues, progress on achieving models with good uncertainty rankings will likely be made by empirical studies specifically focused on the seizure detection task. As already mentioned, another option is to investigate algorithms with an explicit reject (defer) option [32,33,34].

Trust scores almost always outperform SVM confidences. They perform similarly for the DS for classification with a deferral option. The main claim of the article that introduces trust scores is that they produce better uncertainty rankings compared to the classifier itself, at least on low to mid-dimensional datasets [30]. The results on this seizure detection task corroborate this claim. We made an attempt to understand what causes low trust scores in the EEG data (e.g., they are mostly caused by measurement noise), but this proved to be difficult.

An interesting question is how model confidences and trust scores would perform on non-EEG data, such as electrocardiograms, photoplethysmographs, electromyograms and accelerometry data [8,9,49], both for classification with a deferral option and LTF. In a multi-modal setting, one can investigate whether confidence measures can be used to detect which modality works best for detecting seizures for a given patient. The use of different modalities is of interest in some types of seizures, since they can be better detected with the use of an alternative biosignal (e.g., EMG in tonic clonic seizures).

## 5. Conclusions

We have investigated two applications of seizure detection where the classifier has the option not to make a decision. The dataset under study consists of EEG measurements from four behind-the-ear sensors on 54 epileptic patients with focal onset seizures [23]. We expect similar results for other types of seizures (e.g., absence seizures), since the proposed method is being used as a post-processing tool and the type of seizure will not affect (at least not significantly) the conclusions reached in this study. Behind-the-ear measurements can be used for long-term home monitoring outside the hospital. Support vector machine classifiers were already developed in previous work [23]. Prediction confidences are determined by temperature scaling [40] of the SVM output and trust scores [30], which can be calculated independently from the classifier.

In the first application, we investigated the performance gain in the case that part of the data is deferred to a human annotator, who is assumed to annotate perfectly. For both models, a detection sensitivity of approximately 90% (99%) can be achieved when deferring around 10% (40%) of the data. Perfect performance can be reached after deferring 50% of the data. Our results indicate that better deferral strategies, improved classifiers and better confidence measures could provide further improvements.

In the second application, we showed that a common modelling strategy for EEG data, where predictions from several short EEG segments are combined to obtain a final prediction [9,22,23,35], can be improved by filtering out untrustworthy segments. The false detection rate showed a relative decrease of between 21% and 43%, and the detection sensitivity showed a small increase or decrease. Both the positive predictive value and F1-score improved considerably. Filtering only works with trust scores. It does not work with the confidences calculated from the SVM output. This corroborates the results from [30]. Since only one hyper-parameter needs to be optimised, these results suggest that this approach is a relatively straightforward way to improve the performance of a pre-trained classifier, without introducing over-fitting.

## Figures and Tables

**Figure 1 sensors-21-01046-f001:**
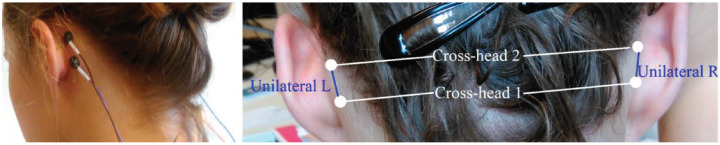
Behind-the-ear electroencephalographic setup. Left panel shows extra behind-the-ear electrodes glued to the skin. Right panel shows bipolar channel derivations. Reproduced with permission from Gu et al. [22].

**Figure 2 sensors-21-01046-f002:**
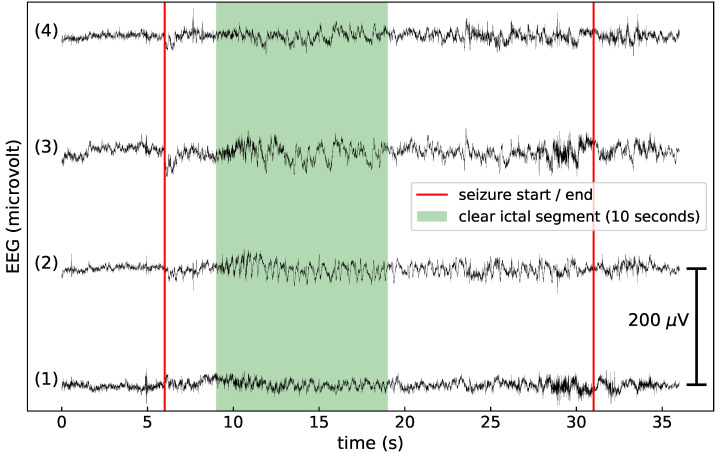
An example of the full seizure and clear ictal labels, used to train the full seizure SVM and clear ictal SVM. (**1**) Crosshead 1, (**2**) crosshead 2, (**3**) unilateral left, (**4**) unilateral right. The full seizure, as annotated by the neurologists, occurs between the vertical red lines. A 10-s EEG segment that contains a clear ictal pattern is also annotated.

**Figure 3 sensors-21-01046-f003:**
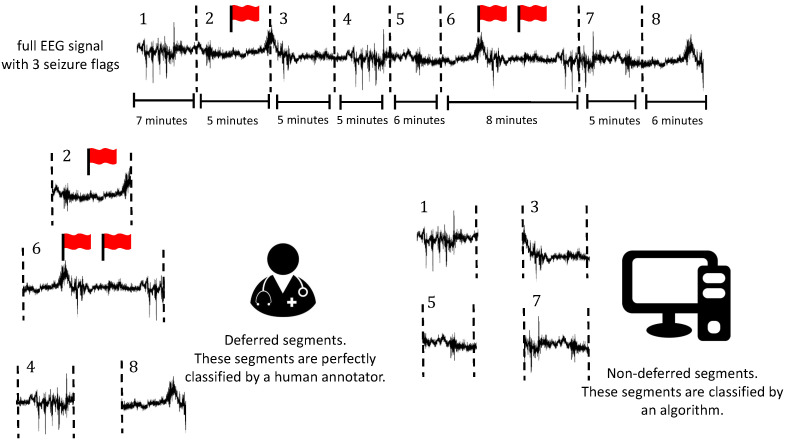
Visualisation of the deferral scheme. EEG segments are deferred to a human annotator or they are classified by an algorithm. The human annotator is assumed to annotate perfectly. Segments that contain seizure flags are always deferred. The minimum length of a segment is 5 min, but they can be longer (e.g., if they contain several seizure flags or are on a boundary). For the example in this figure, segments 2 and 6 are automatically deferred because they contain seizure flags. Segments 4 and 8 are deferred because the classifier is the least confident regarding its classification for these segments. Segments 1, 3, 5 and 7 are classified by the algorithm.

**Figure 4 sensors-21-01046-f004:**
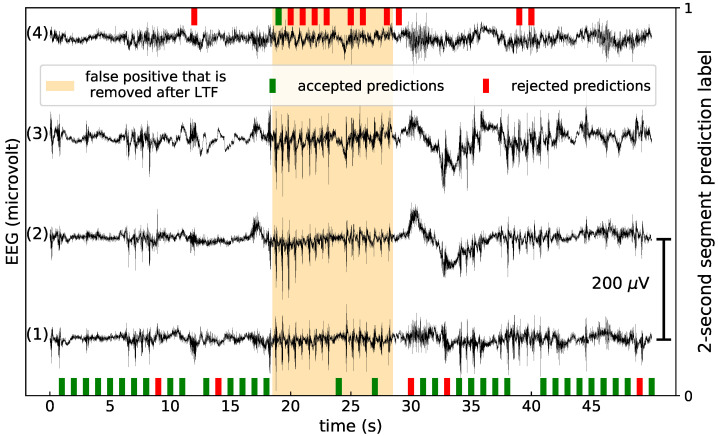
A visualisation of a false positive that was removed after low-trust filtering, for the clear ictal (CI) SVM and a trust model trained on the full seizure (FS) labels. (**1**) Crosshead 1, (**2**) crosshead 2, (**3**) unilateral left, (**4**) unilateral right. If the SVM predicted 0, a bar on the x-axis in the middle of the corresponding 2-s segment is shown. If the SVM predicted 1, a bar on top of the figure is shown. Predictions that are flagged as untrustworthy are shown as red bars; otherwise they are green. The seizure flag that disappeared after low-trust filtering is shown in orange.

**Figure 5 sensors-21-01046-f005:**
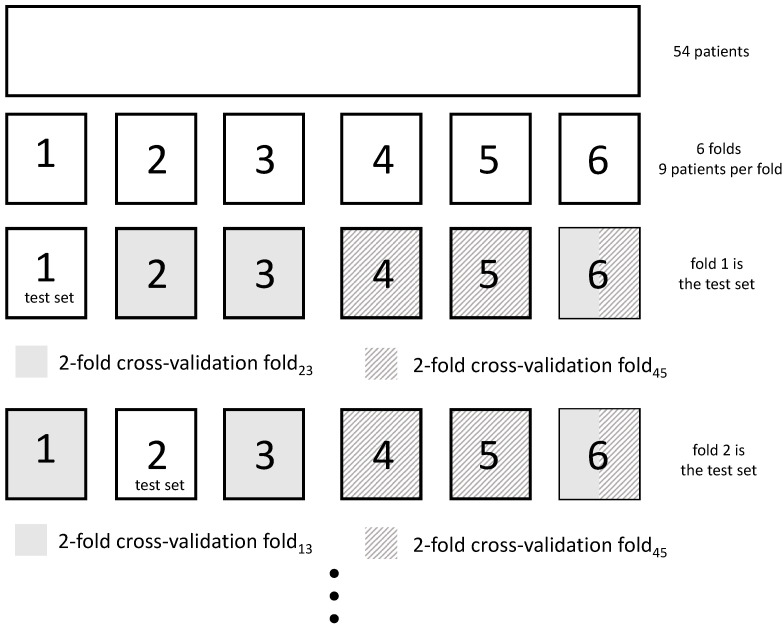
Visualisation of the cross-validation approach for low-trust filtering with trust scores. The 54 patients were divided into 6 folds. Each fold was used as the test set once. The optimal percentage of lowest-trust segments to remove is determined with 2-fold cross-validation. The determination of the folds for cross-validation is visualised when fold 1 or fold 2 is the test set.

**Figure 6 sensors-21-01046-f006:**
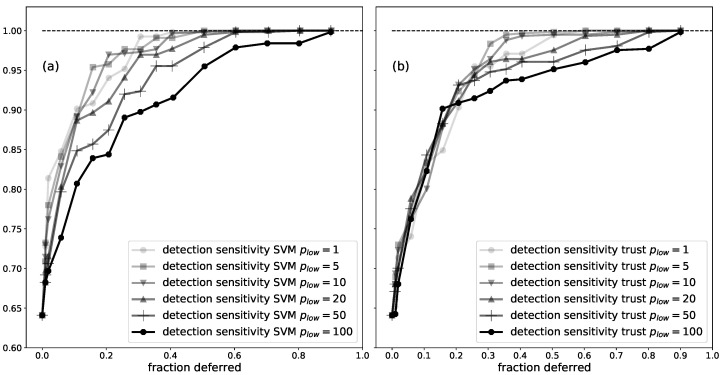
The average detection sensitivity as a function of the fraction of the data that was deferred to a human annotator, for the CI SVM, for different $p_{\mathrm{low}}$. Segments were deferred using (**a**) SVM confidences; (**b**) trust scores from a trust model trained on the FS labels.

**Figure 7 sensors-21-01046-f007:**
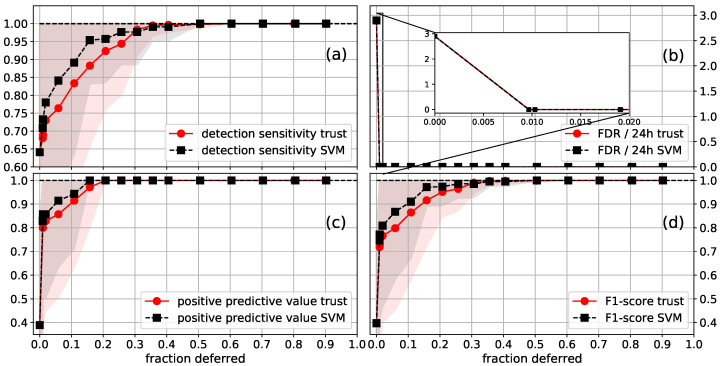
Average (**a**) detection sensitivity; (**b**) FDR/24h; (**c**) PPV; (**d**) F1-score as a function of the fraction of the data that was deferred to a human annotator, for the CI SVM ($p_{\mathrm{low}}=5$). The standard deviation of the performance is shown as a shaded area, with the upper values being capped at one. Segments were deferred using the SVM confidences (SVM) or trust scores (trust) from a trust model trained on the FS labels. The first point with a deferred fraction of larger than zero is the performance when all segments that contain a seizure flag are deferred. The inset plotted in the FDR figure shows that around 1% of the EEG data is contained in segments that contain seizure flags.

**Figure 8 sensors-21-01046-f008:**
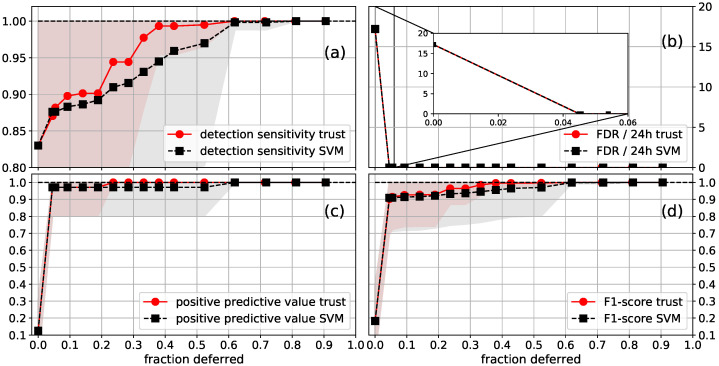
Average (**a**) detection sensitivity; (**b**) FDR/24h; (**c**) PPV; (**d**) F1-score as a function of the fraction of the data that was deferred to a human annotator, for the FS SVM ($p_{\mathrm{low}}=5$). The standard deviation of the performance is shown as a shaded area, with the upper values being capped at one. Segments were deferred using the SVM confidences (SVM) or trust scores (trust) from a trust model trained on the FS labels. The first point with a deferred fraction of larger than zero is the performance when all segments that contain a seizure flag are deferred. The inset plotted in the FDR figure shows that around 4.5% of the EEG data were contained in segments that contained seizure flags.

**Figure 9 sensors-21-01046-f009:**
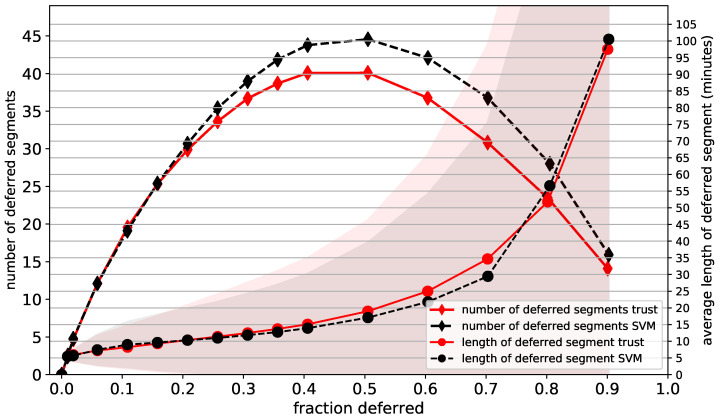
Average number and average length (minutes) of the deferred segments, as functions of the fraction of the data that is deferred to a human annotator, for the CI SVM ($p_{\mathrm{low}}=5$). The standard deviation of the length is shown as a shaded area, with the lower values being capped at zero. Segments were deferred using the SVM confidences (SVM) and trust scores (trust) from a trust model trained on the FS labels.

**Figure 10 sensors-21-01046-f010:**
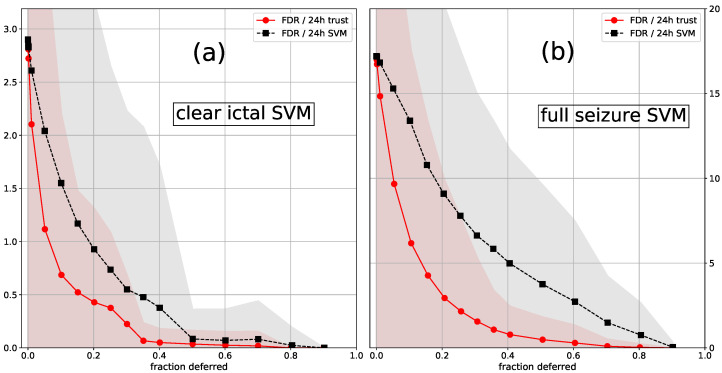
Average FDR/24 h as a function of the fraction of the data that was deferred to a human annotator, for the (**a**) CI SVM; (**b**) FS SVM ($p_{\mathrm{low}}=5$). The standard deviation is shown as a shaded area. In contrast to our main approach, the segments that contain seizure flags were not automatically the first to be deferred. Segments were deferred using the SVM confidences (SVM) and trust scores (trust) from a trust model trained on the FS labels (for both cases).

**Table 1 sensors-21-01046-t001:** Mean (standard deviation) performances of: CI SVM; low-trust filtering (LTF) with the confidences of this SVM (CI SVM conf.); and LTF with trust models trained on CI (trust CI) and FS (trust FS) labels. A statistically significant improvement or degradation in performance compared to the original SVM is denoted by a star. The best result (or not significantly different from the best result) is shown in bold.

Metric/Method	CI SVM	LTF, Trust CI	LTF, CI SVM Conf.	LTF, Trust FS
DS (%)	64.1 (41.5)	63.8 (41.0)	64.1 (41.5)	**71.4** * (38.6)
FDR/24 h	2.9 (5.6)	**1.7** * (3.8)	5.4 * (11.7)	2.3 * (4.7)
PPV (%)	38.9 (38.9)	48.6 * (40.5)	38.7 * (39.0)	50.8 * (40.6)
F1-score (%)	39.7 (34.2)	49.2 * (36.9)	39.4 * (34.3)	52.1 * (36.5)
detection delay (s)	22.1 (13.2)	23.2 * (12.2)	21.9 * (13.0)	21.4 (12.0)

**Table 2 sensors-21-01046-t002:** Median [range] of performances of: CI SVM; LTF with the confidences of this SVM (CI SVM conf.); and LTF with trust models trained on CI (trust CI) and FS (trust FS) labels.

Metric/Method	CI SVM	LTF, Trust CI	LTF, CI SVM Conf.	LTF, Trust FS
DS (%)	100 [0, 100]	83.3 [0, 100]	100 [0, 100]	100 [0, 100]
FDR/24 h	1.2 [0, 31.5]	0.46 [0, 20.5]	2.0 [0, 66.5]	0.58 [0, 24.3]
PPV (%)	23.1 [0, 100]	40.0 [0, 100]	23.1 [0, 100]	37.5 [0, 100]
F1-score (%)	31.6 [0, 100]	50.0 [0, 100]	31.6 [0, 100]	50.0 [0, 100]
detection delay (s)	19.3 [2, 55]	20.3 [3, 56]	19.3 [2, 55]	18.3 [3, 56]

**Table 3 sensors-21-01046-t003:** Mean (standard deviation) performances of: FS SVM; LTF with the confidences of this SVM (FS SVM conf.); and LTF with trust models trained on CI (trust CI) and FS (trust FS) labels. A statistically significant improvement or degradation in performance compared to the original SVM is denoted by a star. The best result (or not significantly different from the best result) is shown in bold.

Metric/Method	FS SVM	LTF, Trust FS	LTF, FS SVM Conf.	LTF, Trust CI
DS (%)	83.0 (30.0)	81.9 (10.6)	83.0 (30.0)	81.6 (32.3)
FDR/24 h	17.2 (21.0)	10.6 * (14.6)	31.6 * (42.4)	9.8 * (14.2)
PPV (%)	12.6 (17.4)	20.3 * (23.9)	12.6 (17.4)	23.8 * (28.3)
F1-score (%)	18.3 (20.3)	27.7 * (27.1)	18.3 (20.3)	30.2 * (29.6)
detection delay (s)	21.6 (17.4)	22.2 * (18.0)	21.6 (17.4)	22.8 * (18.0)

**Table 4 sensors-21-01046-t004:** Median [range] of performances of: FS SVM; LTF with the confidences of this SVM (FS SVM conf.); and LTF with trust models trained on CI (trust CI) and FS (trust FS) labels.

Metric/Method	FS SVM	LTF, Trust FS	LTF, FS SVM Conf.	LTF, Trust CI
DS (%)	100 [0, 100]	100 [0, 100]	100 [0, 100]	100 [0, 100]
FDR/24 h	9.54 [0.26, 98.2]	5.3 [0, 75.8]	16.3 [0.4, 205.7]	4.5 [0, 70.3]
PPV (%)	5.6 [0, 87.5]	12.5 [0, 100]	5.6 [0, 87.5]	12.5 [0, 100]
F1-score (%)	10.5 [0, 87.5]	16.7 [0, 100]	10.5 [0, 87.5]	20.3 [0, 100]
detection delay (s)	16.9 [1, 89]	16.4 [2, 90]	16.9 [1, 89]	16.6 [2, 90]

## Data Availability

The data presented in this study are available on request from W.V.P. or M.D.V. The data are not publicly available due to privacy reasons.

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
