# Peer review of "Classification with a Deferral Option and Low-Trust Filtering for Automated Seizure Detection"

_sensors, 2021, doi:10.3390/s21041046_

Round 1

Reviewer 1 Report

The manuscript tackles two important questions related to automatic seizure detection in the EEG of patients with epilepsy. First, there are already available but there will be also more devices for long term monitoring of seizures in a home monitoring setting based on a reduced electrode set, which will create an increased demand of reliable automatic analysis, and second, how to deal with a large amount of false detections in case an almost perfect sensitivity will be aimed at. Further it has to be mentioned, that in a large amount of studies either overoptimistic results are presented related to the detection accuracy (e.g. by using a very limited data set) or the critical discussion of false detections is neglected (or no suitable accuracy measures are used).

In this study a comprehensive and sufficiently large dataset is used for algorithm evaluation. Code used for the evaluation is made available via an open platform.

The study shows, that classification with a deferral option and low-trust filtering can achieve important improvements in creating an automatic detection and review approach, which can be suited to judge huge datasets as it can be expected from future home recording systems for seizure counting and treatment observation.

Nevertheless some points in the manuscript should be modified:

1. Abstract: The statement “A DS of approximately 90% (99%) can be achieved when automatically classifying around 90% (60%) of the data. Perfect DS can be achieved when automatically classifying 50% of the data.” Is difficult to understand without having read the complete manuscript.

2. The statement: “For focal seizures, no results were achieved that are good enough to be applied in practice.” Is a little bit surprising. Many studies claiming a good detection accuracy, are based on data from presurgical evaluation of epilepsy patients. Only a few examples using more classical or machine learning approaches:

Fürbass F, Ossenblok P, Hartmann M, et al. Prospective multi-center study of an automatic online seizure detection system for epilepsy monitoring units. Clin Neurophysiol

Hopfengärtner R, Kasper BS, Graf W, et al. Automatic seizure detection in long-term scalp EEG using an adaptive thresholding technique: A validation study for clinical routine. Clin

Qaraqe M, Ismail M, Serpedin E, et al. Epileptic seizure onset detection based on EEG and ECG data fusion. Epilepsy Behav 2016;58:48–60.

Gómez C, Arbeláez P, Navarrete M, et al. Automatic seizure detection based on imaged-EEG signals through fully convolutional networks. Sci Rep 2020;10:21833.

Pascual D, Amirshahi A, Aminifar A, et al. EpilepsyGAN: Synthetic Epileptic Brain Activities with Privacy Preservation. IEEE Trans Biomed Eng 2020;9294:1–1.

Or the cited study:

You S, Cho BH, Yook S, et al. Unsupervised automatic seizure detection for focal-onset seizures recorded with behind-the-ear EEG using an anomaly-detecting generative adversarial network. Comput Methods Programs Biomed 2020;193:105472.

Probably the authors wanted express something different: Own work related to absences compared to focal seizures or that automatic seizure detection did not find its way to clinical routine in presurgical evaluation of epilepsy patients up to now.

3. It is described, that for each seizure a 10 second segment with a clear ictal pattern is selected for training. What were criteria in case more than 10 seconds with typical pattern with major artefacts were available ? The first segment, the one with highest frequency or amplitude content ? There exists some tendency of decreasing frequency of the main spectral peak over time, but nevertheless later intervals may be easier to detect just by higher signal amplitudes.

Author Response

We thank the reviewer for the provided comments. Please find attached our response.

Reviewer 2 Report

This is an interesting manuscript that is exceptionally well written, so it has great potential.
My concerns regard some presentation of information as well as the approach of "filtering" out data that reduces performance benchmarks. That must be argued for better or a more honest way of presentation of results should be chosen.

Abstract:
The fact that these high DS can be achieved only on part of the data sounds like cherry picking. It would be better to report the actual DS on the full dataset. Systems are not helpful if they work only for half of the patients.

Introduction:
The information about the study you are currently conducting (lines 31 and following) is not relevant for the Introduction, but you can move it to the methods section.
Line 48-49: what is sufficient for clinical practice?
Line 66: 1670 what?
I have the feeling that sometimes the tense is not correct. In scientific text you should use past tense for your own results but present tense for prior research, unless you use phrases like "authors et al. () did ..." then past tense is ok
It should also not be spoiled in the Introduction what was found in the present research - keep that for the Discussion.

It should be clear in the introduction what is the motivation for the use of only behind-the-ear methods.
Add a section at the end of the Introduction that motivates the use of this system, and the innovative aspects of your work, as well as the aim of the study. This is all implicit in the Introduction but it is nice to have a short and direct statement at the end of the Introduction.

Methods:
line 167: could you give some examples for other reasons?
Line 178-179: what is the typical sample size per patient for this averaging? Can you also provide values for dispersion, not only measures of central tendency?
line 219: that needs to be defined
I am not sure how the optimal parameters cross-validation mixes with the cross-validation approach for low-trust filtering. Is it ensured that there are no ways how to create overfitting of parameters? Maybe this can be explained a bit better. I guess I did not understand the low-trust filtering approach well. What is the purpose of it, to remove ugly EEG segments that reduce the performance benchmarks? IT should be clear what this filtering is about, and what the impact of it is, sounds a bit like systematic cherry picking.
line 247 which distributions did you compare? Those with and without LTF? Please rephrase to make it less confusing.
It is not valid to use a one sided test if you test whether performance is greater OR smaller, then you need a two sided test.

Results
Supplements should only include data that is not relevant to get the message of the paper, i.e. additional information.

Discussion
The meaning and implications of LTF need to be discussed honestly. Does it mean that this will work only for 50% of data, patients etc? Is it then a valid method for clinical practice? It should also stated explicitly that if no data is excluded, the performance is as low as 65 if I get this correctly from figure 5.
It should also be discussed that the method of data recording may be the cause for the need for such a high deferral rate. The quality of the recorded data might not be high enough to reduce from 50% to 5%
I am missing a limitations section in the Discussion, could be an explicit subsection

Author Response

(The authors gave the same response as above.)
